# The Effects of Sampling Depth on Benthic Testate Amoeba Assemblages in Freshwater Lakes: A Case Study in Lake Valdayskoe (the East European Plain)

**Vlad V. Sysoev** [1,*] 🄳, **Andrey N. Tsyganov** [2] 🄳, **Fedor Y. Reshetnikov** [3] **and Yuri A. Mazei** [2,4,5] 🄳

1   Papanin Institute for Biology of Inland Waters, Russian Academy of Science, 152742 Borok, Russia
2   Department of Biology, Lomonosov Moscow State University, Leninskie Gory 1, 119234 Moscow, Russia
3   State Hydrological Institute (SHI), Valday Branch, Pobedy, 2, 175400 Valdai, Russia
4   A.N. Severtsov Institute of Ecology and Evolution, Russian Academy of Sciences, Leninskiy Ave. 33, 117071 Moscow, Russia
5   Faculty of Biology, Shenzhen MSU-BIT University, Shenzhen 518100, China
*   Correspondence: vladimirsysoev@rambler.ru; Tel.: +7-915-344-52-75

**Abstract:** Testate amoebae are widely used as proxies in paleoecological reconstructions of lacustrine environments; however, our knowledge on their distribution along depth gradients are limited. This study investigates the distribution of benthic testate amoebae along a sampling depth gradient (0 to 57 m) and related environmental characteristics in Lake Valdayskoe, Russia. In total, 101 species belonging to twenty-one genera were identified. Four types of testate amoeba assemblages (littoral, sublittoral, bottom slope and profundal) were distinguished that corresponded well to the bottom zones of the lake. The results of redundancy analysis indicated that sampling depth, temperature, pH and bottom inclination significantly explained 40.2% of the total variance in the species composition. Temperature and sampling depth had the largest individual contributions of 19.2 and 7.4% ($p < 0.001$), respectively. The minimal values of species diversity were observed on the littoral and at the lower boundary of the thermocline. We estimated depth optima and ranges for the species with high occurrences and distinguish stenobathic and eurybathic species. These data might improve the interpretations of paleoecological records of subfossil testate amoeba assemblages in lacustrine surface sediments and serve as basis for the development of a transfer function for reconstruction of lake depths.

**Keywords:** lake; depth gradient; littoral; sublittoral; bottom slope; profundal; protists; dominance structure; ecological preference; lake sediment

## 1. Introduction

Testate amoebae represent a polyphyletic group of unicellular eukaryotic organisms characterized by decay-resistant and morphologically distinguished shells (or tests) living worldwide in freshwaters and playing an important role in benthic assemblages where they are usually the top predators among the microorganisms [1–3]. Testate amoebae are widely used as indicators (proxies) of eutrophication level, water temperature, water depth, etc. in the paleolimnological reconstructions along with diatoms, crustaceans, chironomids and pollen [4–8]. The ability and practicability of using testate amoeba as a proxy for the reconstruction of lake dynamics and palaeoecological analysis is linked to their high reproduction rate and quick response to environmental changes [9] as well as to the wide distribution and good preservation of shells in lake sediments since the beginning of the Holocene [10].

Studies on benthic assemblages of testate amoebae have been done in many regions such as North and Latin America [11–13], Europe [14,15], China [16], and Russia [17]. In the last decades there was a number of papers that described the relationships between

various abiotic factors, e.g., pH, temperature, oxygen content, eutrophication and sediment characteristics, and the species structure of testate amoeba assemblages [12,13,18–21]. The sampling depth seems to be one of the most significant factors, influencing the species structure of testate amoeba assemblages since the key environmental characteristics (e.g., sediment properties, organic matter content, acidity, temperature and eutrophication) change considerably along depth gradients [14,17]. At the same time, the depth in the freshwater bodies is one of the most important characteristics for the paleoclimatic and paleoecological reconstructions [22,23].

The effects of sampling depth on species structure of testate amoeba assemblages including a comparison of phytal, littoral, and profundal habitats were demonstrated in several publications [24–27] along with the possibility to use testate amoebae as bioindicators of depth in the freshwater bodies [17] and seas [28]. The aim of this work is to investigate the effects of sampling depth on species structure of testate amoeba assemblages in surface sediment samples along a depth gradient in a large freshwater lake, in order to evaluate the possibility of using testate amoebae in paleolimnological studies as a proxy for reconstruction of lake depth changes during the Holocene. We hypothesized that sampling depth will affect diversity and species structure of testate amoeba assemblages in the bottom surface sediments that results in well-defined ecological preferences of testate amoeba taxa. Understanding the patterns of testate amoeba assemblages along the depth gradient is also crucial for implementing an efficient sampling design to assess anthropogenic impact on freshwater ecosystems.

## 2. Materials and Methods

### 2.1. Study Site

Lake Valdayskoe (57°59′00″ N 33°18′00″ E, 192 m a.s.l.) is located on the Valdai Upland on the territory of the Valdai National Park in the Novgorod Region (Figure 1a). The total area of the lake without islands is 19.7 km², the lake size is 40 km long and 32 km wide. The basin of the lake belongs to a mixed accumulative-subsidence type. The lake is located in the terminal moraine zone of the Krestetsky stage of the Valdai glaciation (13,000 years ago). The cone-shaped form of the lake basin with the depths up to 60 m and steep lake basin slopes indicate that during the last glacial period, the lake existed in the form of dead ice. Then, the ice gradually melted and the material left on it was partially carried out of the current catchment area of the lake and partially lay on the lake bottom creating islands. The lake basin consists of four clearly defined zones—littoral, sublittoral, bottom slope, and profundal [29].

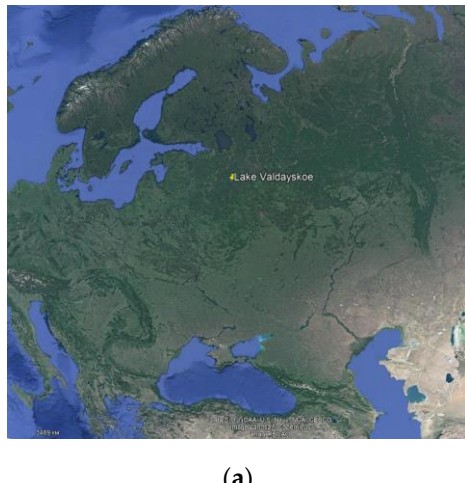

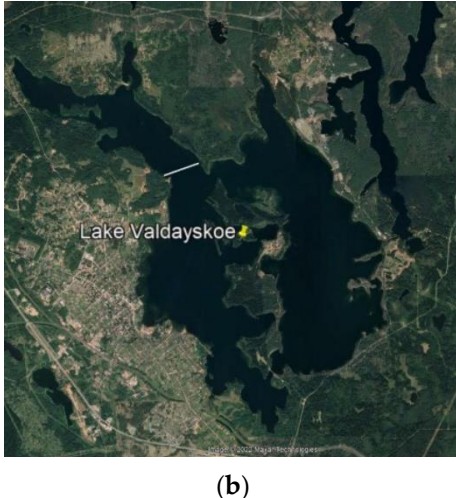

| (a) | (b) |

**Figure 1.** Map of the study region (**a**) with the location of the Lake Valdayskoe and the study site (**b**) with the location of the transect shown by a white line (https://www.google.ru/maps, accessed on 1 October 2022).

The trophic status of the lake water identified by the concentration of chlorophyll "a" is α-mesotrophic [29], whereas the benthic biomass indicates various degrees of eutrophic conditions [30]: the littoral is hypereutrophic, the sublittoral is α-mesotrophic and the trophic status of the profundal zone decreases to β-oligotrophic. The near-bottom water contains a sufficient amount of oxygen so that even at a depth of 30 m the oxygen content is up to 5 mg m$^{-3}$ [29]. Lake Valdayskoe can be subdivided into three water layers with different temperature, oxygen concentration and levels of illumination. The epilimnion is the upper water layer (depth from 0 to 6 m), which is characterized by high light intensity and a significant concentration of dissolved oxygen; wind and waves constantly stir the water in this layer. The second water layer (at the depths of 6 to 15 m) is represented by the thermocline and appears only in the ice-free period. This layer is characterized by a considerable decline in temperature (the gradient is more than 2 °C per meter in the upper part and between 0.5 and 2 °C per meter in the lower). The third layer, hypolimnion, is located at the depths greater than 15 m and characterized by low temperatures and weak water mixing [29].

### 2.2. Field Sampling

Lacustrine bottom sediments were sampled along a transect (starting point 58.002589 °N, 33.264995 °E; end point 58.004447 °N, 33.274646 °E) in July 2021 (Figure 1b). The transect length was 551 m; the depths along the transect ranged from 0 to 57 m. The profile of the lake basin along the transect was done using the River Ray sub-bottom profiler. Samples of bottom surface sediments were taken from a boat with a Birge-Eckman dredger (Figure S1) with sampling depth intervals of 3 m. Three repeated samples were taken at each site. The top 3 cm of the surface sediments, of a total volume 10–20 cm$^3$, were collected for testate amoeba analysis and fixed with 4% formalin. In total, 20 sampling sites were investigated and 60 samples were collected. The surface sediments were described in the field and classified following Nedogarko [29]. Sampling depths and water temperature were measured by a lot and a thermometer depth gauge (T OTT KL 010 TM), respectively, during the sample collection. Samples of near-bottom water were collected using a bathometer for pH measurements (HM Digital PH-80).

### 2.3. Testatea Amoeba Analysis

For testate amoeba analysis, 3–5 mL of the sediments were mixed with 1–3 mL of glycerin and investigated using a light microscope Olympus CX41 at a magnification of ×200. All encountered shells were counted and identified to a minimal total of 150 individuals per sample which, overall, resulted in more than 450 individuals per site (depth). These counts are considered sufficient for a reliable description of the species diversity and structure of testate amoeba assemblages [31]. The following identification guides were used Todorov & Bankov [32] and Mazei & Tsyganov [33].

### 2.4. Data Analysis

Numerical calculations and statistical analyses were performed in the R language environment [34] with the supplementary packages 'vegan' [35] and 'rioja' [36]. To analyze the distribution of species diversity along the depth gradient, we calculated the total species number per sampling depth (three samples combined), the mean number of species in three samples per depth, and Shannon and Simpson diversity indices [37].

Shannon diversity index (*H*) was calculated following the formula:

$$H = - \sum [(p_i) * (\ln p_i)] \tag{1}$$

where $p_i$—proportion of *i*-th species to the total number of shells in the community.

The Simpson diversity index (*D*) was calculated following the formula:

$$D = 1 - \sum \left( \frac{n}{N} \right) 2, \qquad (2)$$

where *n* is the number of shells of a given species; and *N* is the total number of shells in a community.

The effect of the sampling depth, temperature, pH, and angle of the bottom on the species structure of testate amoeba assemblages was evaluated with redundancy analysis (RDA; 'rda' function in the 'vegan' package). To group samples according to similarities in species composition along the water depth transects, a constrained cluster analysis with an incremental sum of squares [38] was performed ('chclust' function in the 'rioja' package). Indicator species for each cluster were identified following the IndVal approach with 'indval' function in the package "labdsv" [39].

Relative abundance of genera (*OR*) and species (*S*) were calculated following the formulas:

$$OR_x = \frac{N_x}{\sum N_{total}}, \qquad (3)$$

where $OR_x$ is the relative abundance of the genus *x*, $N_x$ is the number of shells of the genus *x* in the sample, $N_{total}$ is the total number of shells in the sample;
and

$$S_x = \frac{M_x}{\sum M_{total}}, \qquad (4)$$

where $S_x$ is the relative abundance of the species *x*, $M_x$ is the number of counted shells of the species *x* in the sample, $M_{total}$ is the total number of tests in the sample.

The total (*TS*) and the mean (*MS*) number of species at the sampling depth were calculated following the formula:

$$TS = N_1 + N_2 + N_3, \qquad (5)$$

where *N* is the number of species in a sample, 1, 2, and 3 are the sample numbers at a sampling depth,
and

$$MN = (N_1 + N_2 + N_3)/3, \qquad (6)$$

where *N* is the number of species in a sample, and 1, 2, and 3 are the sample numbers at a sampling depth.

The dominance curves [40] were built by ranking the species by their relative abundance in decreasing order.

The bottom inclination angle was calculated based on basin profile data for intervals between underwater terraces or between bottom zones with significantly different inclination angles. The intervals 0–34, 34–111, 170–331, 332–375, 430–551 m from the shore were identified. The inclination angle was calculated using the following formula:

$$tg(\alpha) = a/b, \qquad (7)$$

where *tg(α)* is the tangent of the angle of inclination, *a* is the depth difference within the interval and *b* is the length of the interval.

## 3. Results

### 3.1. Characteristics of the Transect

The littoral zone of the transect extended for about 80 m from the shore and consisted of two parts (Figure 2). The upper part (30 m long) started from the water edge so that the depth increased from 0 to 0.7 m with an angle of inclination about 2°. The lower part (50 m long) was characterized by a greater angle of 6°, whilst the depth increased from

0.7 to 6 m. The sublittoral zone was located in the depth range from 6 to 12 m and was mainly represented by a flat terrace (60 m long) at the depth 9 m. The bottom slope zone started around 170 m from the shore and was located at depths from 12 to 30 m. The upper part of the bottom slope (to the depth of 18 m) was characterized by an angle of inclination of about 4° and the lower part (18–26 m) had a greater angle of 10°. The bottom slope terminated with a deep-water terrace (60 m long) at depths of 26–30 m. The profundal zone declined to the maximal depths of 57 m at the angle of 14°.

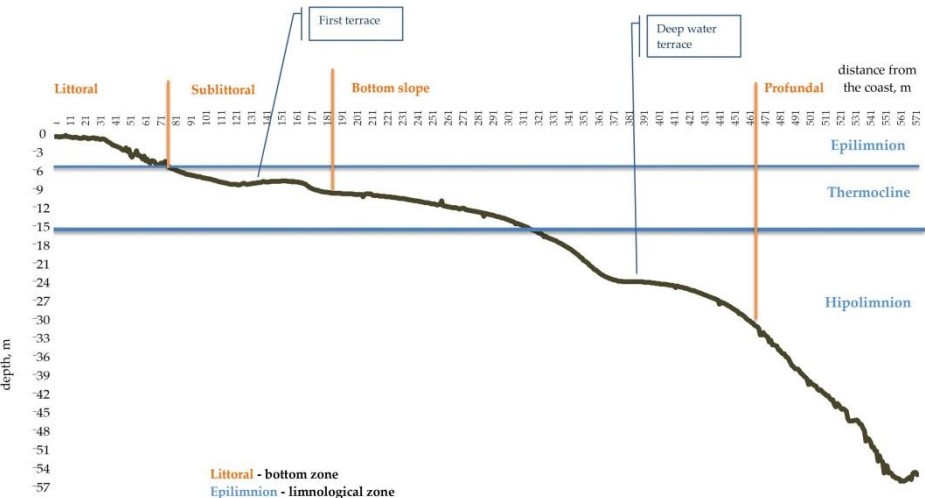

**Figure 2.** The basin profile of Lake Valdayskoe.

The water temperature at the bottom changed from 25.1 °C at the depth of 0 m to 5.9 °C at 57 m (Figure 3a). The thermocline was located at depths from 6 to 12 m. The most considerable temperature drop of 11.8 °C was observed at depths from 6 to 9 m. The near-bottom water pH decreased with depth from 8.9 at 0 m to 7.2 at 57 m (Figure 3b). The surface sediments at the depths of 0–3 m were mainly formed by sand with a small amount of sapropel and silted sand. Starting from the depths of 6 m, the proportion of sapropel in the surface sediments increased (the most common surface sediments were sapropel with a small amount of sand) to the pure sapropel substrates deeper than 9 m. The vegetation in the littoral zone to the depth of 3 m was represented by *Sagittaria sagittifolia* L. (1753) (arrow leaf), *Nymphaea* sp. (water lily), and *Ceratophyllum demersum* L., (1753) (hornwort). At the depths of 3 and 6 m, numerous colonies of *Dreissena* sp. were observed (Table S1).

### 3.2. General Characteristics of Testate Amoeba Assemblages

In the samples, 9667 shells belonging to 21 genera and 101 species were identified (Table 1 and Table S2). The genus *Difflugia* was characterized by the greatest species number (44 species) and abundance (4817 shells). The second was *Centropyxis* with a total of 18 species and 1803 shells, which was followed by the genus *Cylindrifflugia* [41] with five species and 1567 shells, the genus *Netzelia* with two species and 619 shells, and the genus *Arcella* with four species and 295 shells. Other genera found in Lake Valdaiskoe were represented usually by one or two species, up to a maximum of five. The most abundant species were *Difflugia oblonga* (13%, here and further of the total number of counted shells) and *Cylindrifflugia elegans* (12%). The number of identified shells of *Centropyxis cassis*, *C. aculeata*, *Difflugia petricola*, *D. penardi*, *D. linearis*, *D. pulex*, *D. lithophila*, *Netzelia gramen*, and *N. oviformis* was more than 3%. Eleven species were represented by one or two specimens only: *Centropyxis marsupiformis*, *Cryptodifflugia compressa*, *Cyphoderia calceolus*, *C. trochus amphoralis*, *Difflugia angulostoma*, *D. cylindrus*, *D. dujardini*, *Lagenodifflugia bryophila*, *Nebela parvula*, *Oopyxis cyclostome*, and *Trigonopyxis arcula*.

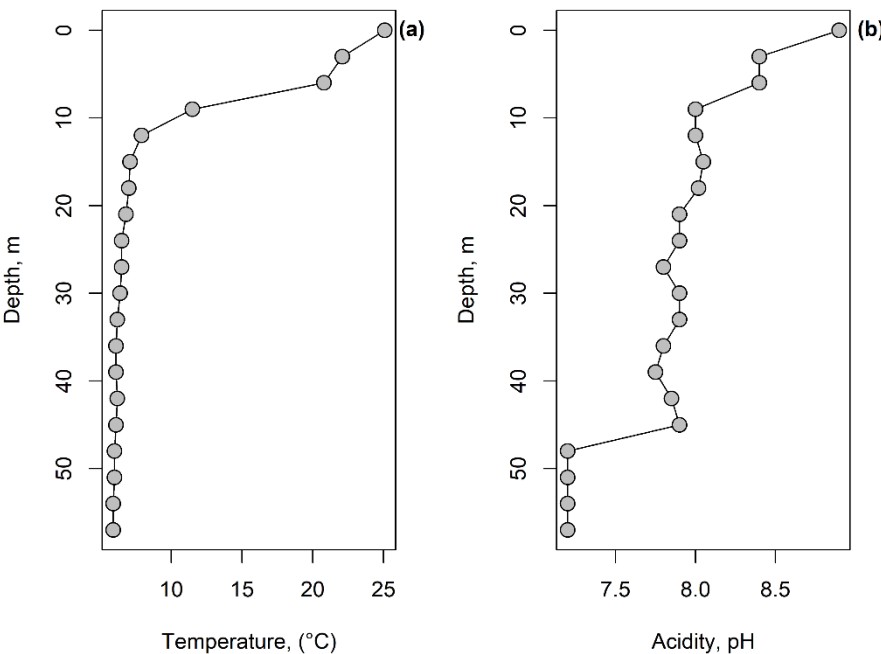

**Figure 3.** Variation of temperature (°C) (**a**) and pH (**b**) of near-bottom water along the depth gradient.

**Table 1.** The list of testate amoeba genera with the number of species and counts along the transect.

| Genus. | Number of Species | Number of Shells | Species Share, % | Shell Share, % |
|---|---|---|---|---|
| *Arcella* Ehrenberg, 1830 | 4 | 295 | 4 | 3.1% |
| *Centropyxis* Stein, 1857 | 18 | 1803 | 18 | 18.7% |
| *Cryptodifflugia* Penard, 1890 | 5 | 10 | 5 | 0.1% |
| *Cyclopyxix* Deflandre, 1929 | 3 | 60 | 3 | 0.6% |
| *Cylindrifflugia* González-Miguéns et al., 2022 | 5 | 1567 | 5 | 16.2% |
| *Cyphoderia* Schlumberger, 1845 | 3 | 15 | 3 | 0.2% |
| *Difflugia* Leclerc, 1815 | 44 | 4817 | 44 | 49.8% |
| *Erugomicula* Nasser et al., 2022 [42] | 1 | 6 | 1 | 0.1% |
| *Galeripora* González-Miguéns et al., 2022 | 2 | 142 | 2 | 1.5% |
| *Golemanskia* González-Miguéns et al., 2022 | 1 | 91 | 1 | 0.9% |
| *Hyalosphenia* Stein, 1857 | 1 | 6 | 1 | 0.1% |
| *Lagenodiffludia* Medioli et Scott, 1983 | 1 | 1 | 1 | 0.0% |
| *Lesquereusia* Schlumberger, 1845 | 1 | 39 | 1 | 0.4% |
| *Nebela* Leidy, 1874 | 2 | 3 | 2 | 0.0% |
| *Netzelia* Ogden, 1979 | 2 | 619 | 2 | 6.4% |
| *Oopyxis* Jung, 1942 | 1 | 1 | 1 | 0.0% |
| *Paraqudrula* Deflandre, 1932 | 1 | 2 | 1 | 0.0% |
| *Pontigulasia* Rhumbler, 1896 | 3 | 113 | 3 | 1.2% |
| *Pseudodifflugia* Schlumberger, 1845 | 1 | 7 | 1 | 0.1% |
| *Trigonopyxis* Penard, 1912 | 1 | 1 | 1 | 0.0% |
| *Zivkovicia* Ogden, 1987 | 1 | 69 | 1 | 0.7% |
| Total | 101 | 9667 | | |

The number of species per sample varied from 17 to 38. The mean number of species per sample was 25.57 ± 4.7 (SD). Thirteen species were found in more than 75% of samples: *Difflugia oblonga* (100%), *Cylindrifflugia elegans* (100%), *Centropyxis cassis* (98%), *Difflugia penardi* (93%), *D. petricola* (93%), *Cylindrifflugia acuminata* (88%), *Difflugia lithophila* (85%), *D. minuta* (78%), *D. pulex* (78%), *Netzelia oviformis* (78%), *D. linearis* (77%), *D. longicollis* (75%), and *Arcella hemisphaerica* (75%) (Table S3).

*3.3. Variation in Species Structure of Testate Amoeba Assemblages along the Depth Gradient*

Cluster analysis allowed us to identify four types of assemblages that correspond to the lake bottom zones (Figure 4):

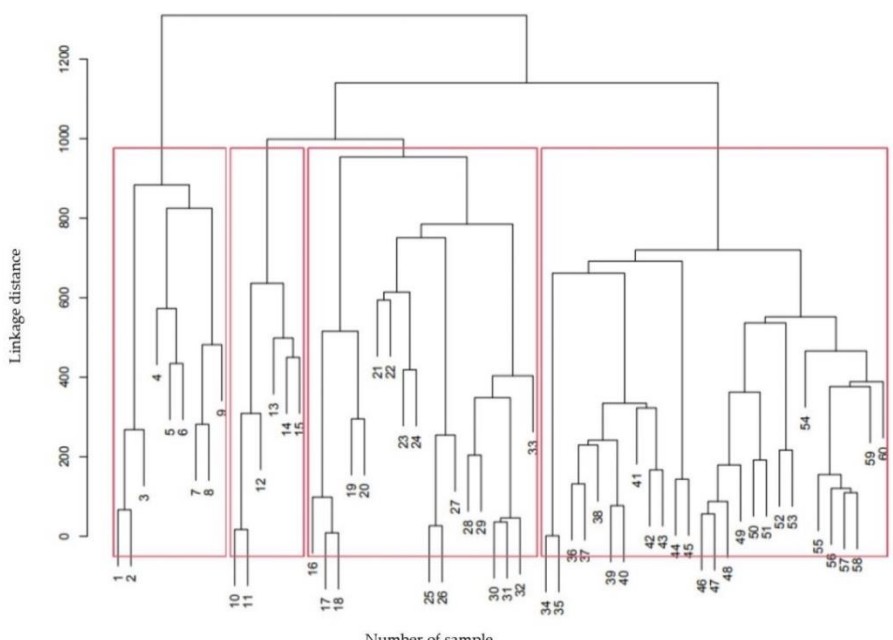

**Figure 4.** Main types of testate amoeba assemblage along the sampling depth gradient based on the results of constrained cluster analysis (the numbers on the dendrogram correspond to the number of samples in Table S1). Red boxes on the plot mark the main types of testate amoeba assemblages.

1. **Littoral assemblage (LA)** (depth 0–6 m). Environment conditions: temperature 25.1–20.8 °C, pH 8.9–8.4, macrophytes and mollusk shells on the bottom, sand and plant residues in the surface sediments (Table S1). At these depths, 74 species belonging to 14 genera were found (Table S2). The most abundant genera were *Centropyxis* (41% herein and after the total shells counts in the zone), *Difflugia* (29%), *Cylindrifflugia* (13.6%), *Arcella* (7.4%) and *Galeripora* [41] (4.5%) (Table S4). The species with the maximum indicator value for the assemblage were *Centropixys discoides*, *C. aculeata*, *Galeripora discoides*, *Cylindrifflugia acuminata,* and *Difflugia lacustris* (Table 2). An analysis of the structure of the LA showed the presence of one dominant species *Centropyxis aculeata* (19%, hereafter the relative abundance in the community), and three subdominants: *C. discoides* (9%), *Difflugia linearis* (6%), and *Cylindrifflugia acuminata* (5.3%); the proportion of the other species did not exceed 5% (Figure 5a). Shannon's diversity index was 3.24.

2. **Sublittoral assemblage (SA)** (depth 9–12 m). Environmental conditions: temperature 7.9 °C, pH 8, presence of plant residues in the surface sediments. The SA consists of 65 species belonging to 11 genera (Table S2). The SA was placed in the thermocline. Genera *Difflugia* (56.5%), *Centropyxis* (15.3%), and *Cylindrifflugia* (14%) were characterized by the greatest relative abundance (Table S4). The species with the maximum indicator value in the SA were *Pontigulasia rhumbleri*, *Difflugia claviformis*, *D. urceolata*, *Golemanskia viscidula* [41], *D. petricola*, and *Zivcovicia spectabilis* (Table 2. The SA was dominated by species *Difflugia petricola* (17.3%) with three subdominant species *D. oblonga* (13.1%), *Cylindrifflugia elegans* (12%), and *Centropyxis cassis* (6.7%). The proportion of other species did not exceed 4% (Figure 5b). Shannon's diversity index was 3.26.

3. **Bottom slope assemblage (BA)** (depth 15–30 m). Environmental conditions–temperature 7.1–6.4 °C, pH 7.9–8.0. The BA was located under the thermocline. In this zone, 67 species belonging to 17 genera were found (Table S2). Genera *Difflugia* (64.2%),

*Cylindrifflugia* (16.3%), and *Centropyxis* (11.6%) had the highest relative abundance (Table S4). The species with the maximum indicator value in the BA were *Difflugia oblonga* and *D. lithophila* (Table 2). The community structure of the BA is characterized by one dominant *D. oblonga* (24%) and five subdominats *Cylindrifflugia elegans* (12%), *D. petricola* (10%), *D. penardi* (7%), *Centropyxis cassis* (7%), and *D. lithophila* (4.6%). The proportion of other species was 4% or less (Figure 5c). Shannon's diversity index was 2.98.

4.　**Profundal assemblages (PA)** (depth 33–57 m). Environmental conditions: temperature 5.9–6.2 °C, pH 7.9–7.2. 87 species belonging to 20 genera were found at this depth (Table S2). Genera *Difflugia* (45.2%), *Cylindrifflugia* (17.6%), *Centropyxis* (16.9%), and *Netzelia* (12.5%) had the highest relative abundance (Table S4). Species with the maximum indicator value in the PA were *Netzelia oviformis*, *N. gramen*, *Difflugia minuta*, and *D. pristis* (Table 2). The community structure of the PA was characterized by three dominant species: *Cylindrifflugia elegans* (14.4%), *Centropyxis cassis* (9.5%), and *Difflugia oblonga* (9.2%) as well as five subdominants *D. penardi* (8.1%), *Netzelia gramen* (6.3%), *N. oviformis* (6.3%), *D. petricola* (4.8%), and *D. linearis* (4.3%) (Figure 5d). Shannon's diversity index was 3.15.

**Table 2.** Indicator species of four assemblages along the sampling depth gradient.

| Assemblage | Taxa | Indicator Value | Probability |
|---|---|---|---|
| **Littoral assemblage** | *Centropyxis discoides* | 0.7522 | 0.001 |
| | *Galeripora discoides* | 0.6857 | 0.001 |
| | *Centropyxis aculeata* | 0.6814 | 0.001 |
| | *Cylindrifflugia acuminata* | 0.5992 | 0.006 |
| | *Difflugia lacustris* | 0.5065 | 0.012 |
| | *Arcella hemisphaerica* | 0.4945 | 0.007 |
| | *Cylindrifflugia lanceolata* | 0.4745 | 0.013 |
| | *Difflugia linearis* | 0.4642 | 0.01 |
| | *Centropyxis ecornis* | 0.4148 | 0.017 |
| | *Arcella vulgaris* | 0.4074 | 0.017 |
| | *Centropyxis cassis spinifera* | 0.4058 | 0.04 |
| | *Difflugia rubescens* | 0.3865 | 0.015 |
| | *Difflugia corona* | 0.3616 | 0.011 |
| | *Arcella rotundata* | 0.3296 | 0.02 |
| **Sub-littoral assemblage** | *Pontigulasia rhumbleri* | 0.75 | 0.001 |
| | *Difflugia claviformis* | 0.6696 | 0.001 |
| | *Difflugia urceolata* | 0.5709 | 0.002 |
| | *Golemanskia viscidula* | 0.5294 | 0.001 |
| | *Difflugia petricola* | 0.5145 | 0.001 |
| | *Zivkovicia spectabilis* | 0.5046 | 0.005 |
| | *Pontigulasia compressa* | 0.477 | 0.002 |
| | *Pontigulasia spiralis* | 0.4091 | 0.005 |
| | *Difflugia longicollis* | 0.3913 | 0.027 |
| | *Difflugia avellana* | 0.3841 | 0.013 |
| | *Difflugia sinuata* | 0.3 | 0.021 |
| | *Erugomicula bidens* | 0.2927 | 0.02 |
| **Bottom slope assemblage** | *Difflugia oblonga* | 0.4709 | 0.001 |
| | *Difflugia lithophila* | 0.4018 | 0.012 |
| | *Cyphoderia ampulla* | 0.2917 | 0.04 |
| **Profundal assemblage** | *Netzelia gramen* | 0.6727 | 0.001 |
| | *Difflugia minuta* | 0.5354 | 0.001 |
| | *Difflugia pristis* | 0.5349 | 0.001 |
| | *Difflugia penardi* | 0.3925 | 0.001 |
| | *Cylindrifflugia elegans* | 0.3277 | 0.009 |
| | *Netzelia oviformis* | 0.682 | 0.001 |

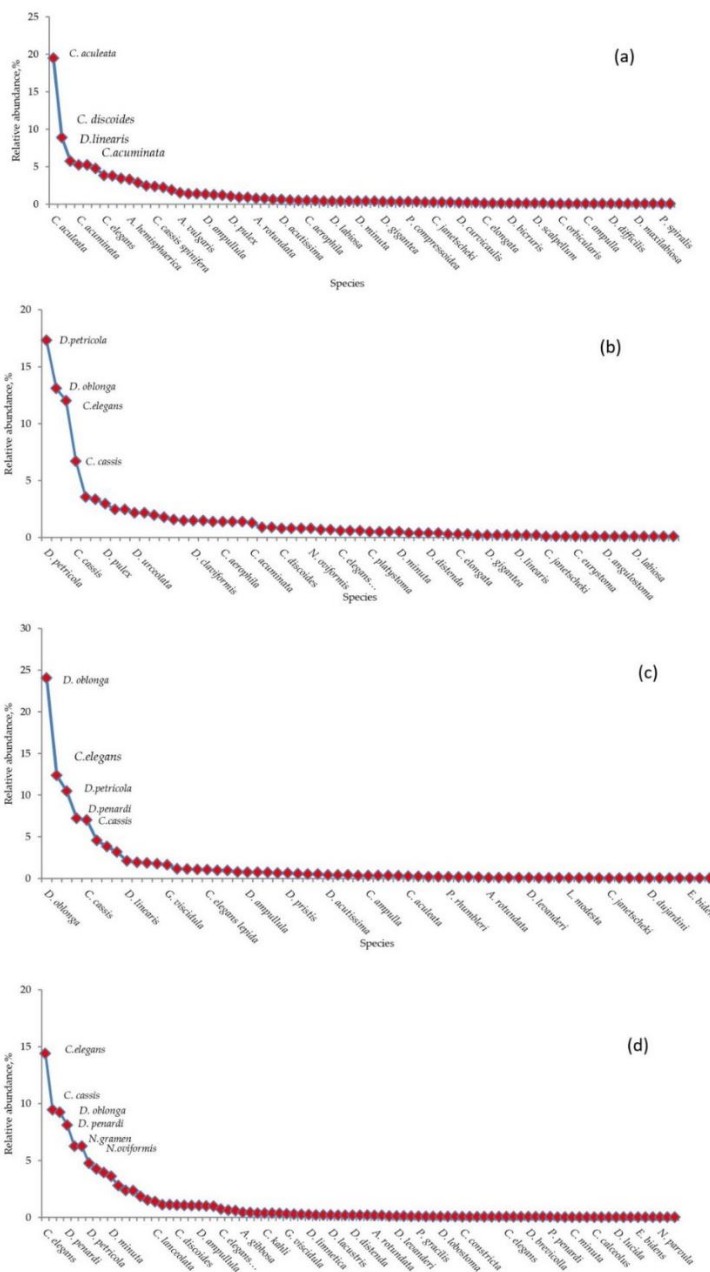

**Figure 5.** Dominance curve of the littoral assemblage (**a**), the sublittoral assemblage (**b**), the bottom slope assemblage (**c**), and the profundal assemblage (**d**).

The results of RDA indicated that four explanatory variables (sampling depth, temperature, pH, and bottom inclination) explained together 40.2% of the total variance in the species composition (*pseudo-F* = 10.9, *p* < 0.001). Variance partitioning (Figure 6) showed that temperature and sampling depth had the largest individual fractions of 19.2 and 7.4% (*p* < 0.001), respectively. The individual contributions of pH and bottom inclination were significant (*p* < 0.002 and *p* < 0.041, respectively) but overall negligible (<1.5%). Surprisingly the total proportion of the explained variance shared among the predictors was relatively low 11.6% which indicates their specific contribution to species composition of testate amoeba assemblages.

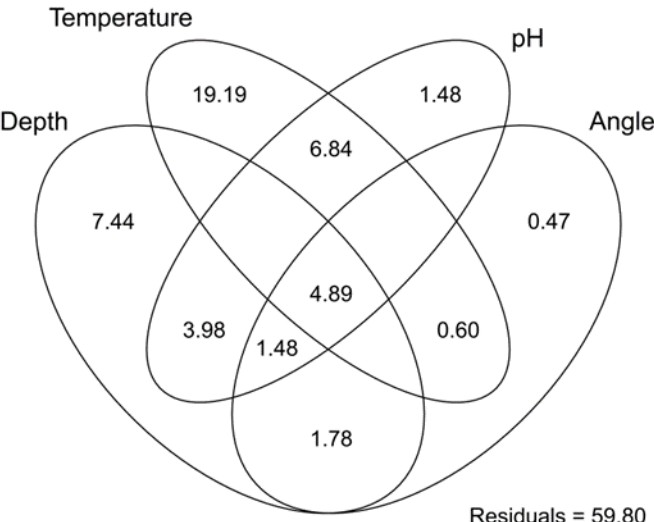

**Figure 6.** Venn diagram of variance partitioning analysis illustrating the percentage of variance in the testate amoeba assemblage data explained by sampling depth, temperature, pH, and lakebed angle of inclination. Values show percentages of the variance explained by each variable and joined effects of their interaction (computed from adjusted $R^2$). The analysis is based on the results of redundancy analysis (relative abundance data, covariance matrix). The significance of the testable fractions is given in Table 2. Values less than zero are not shown.

*3.4. Relative Abundance of Testate Amoeba Genera along the Depth Gradient*

　　The relative abundance of genera changed along the depth gradient, which demonstrated the differences in genus responses to the water depth (Figure 7). The relative abundance of the genus *Difflugia* increased with depth from the littoral to the beginning of the bottom slope (15 m depth), where it reached a maximum value of 74% (the proportion to the total number of tests found at this depth). From 18 m and deeper, on the bottom slope and in the profundal zone, the relative abundance of the genus *Difflugia* decreased. Two peaks in relative abundance were found at depths of 27–30 and 42 m, and two drops were detected at the depths of 33 and 48 m. The relative abundance of the genus *Centropyxis* was maximal on the littoral where it reached 60% at the depth of 0 m. In the sublittoral zone and deeper, the relative abundance strongly decreased and varied from 3 to 25% with the lowest value at 15 m. The relative abundance of the genus *Cylindrifflugia* was a minimum of 9% at the depth of 6 m and then slightly increased in the sublittoral and bottom slope to 22% at the depth of 27 m and varied from 14% to 23% at greater depths. The relative abundance of the genus *Netzelia* did not exceed 5% to the depth of 30 m and then increased to 16% in the profundal zone at the depths of 33 m and varied between 11 and 17% deeper. The maximal relative abundance (11%) of the genus *Arcella* was registered on the littoral. On the sublittoral and at the bottom slope the relative abundance of this genus varied between 0 and 4%, but it increased to 8% in the profundal zone at the depths of 45 and 48 m (Figure 7).

　　The relative abundance of genus *Galeripora* was maximal (8.6%) on the littoral at 6 m and then the shells of the genus almost disappear to the depth of 18 m (Figure S2a). The genus *Golemanskia* [41] had an apparent peak (6.4%) of relative abundance at the bottom slope (15 m) and at the other depths, the relative abundance was quite low and did not exceed 4% (Figure S2a). Changes in the relative abundance of the genera *Pontigulasia*, *Zivkovicia*, and *Cyclopyxis* along the depth gradient were similar (Figure S2a). The highest relative abundances (3–4%) of these genera were on the sublittoral at depths of 9 and 12 m and ranged from 0.3–1.3% at the other depths.

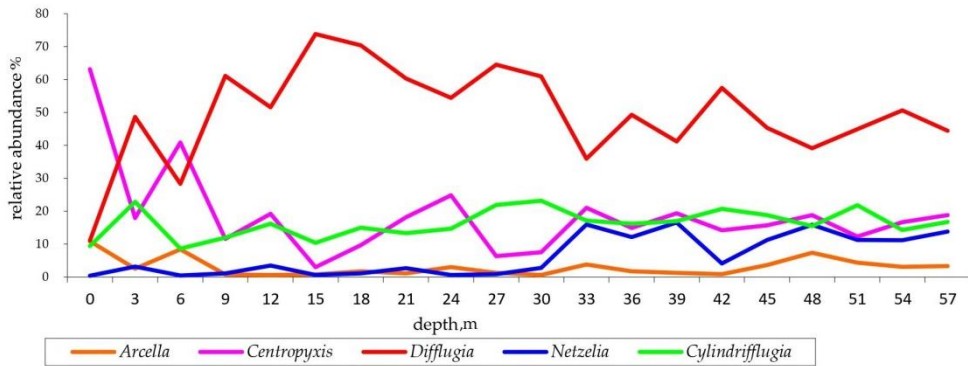

**Figure 7.** Relative abundance (%) of the genus *Arcella*, *Centropyxis*, *Difflugia, Netzelia* and *Cylindrifflugia* along the depth gradient.

The genus *Lesquereusia* was mostly found on the littoral and sublittoral (depths of 3, 6, and 9 m) where it reached the maximal relative abundance of 0.9–1.5%. In the bottom slope at the depths between 12 and 33 m, *Lesquereusia* appeared in small amounts and a small peak of relative abundance was registered in the profundal zone at the depths greater than 33 m (Figure S2b). Specimens of the genera *Cyphoderia* were evenly distributed over all depths (Figure S2b). A small number of *Cryptodifflugia* shells appeared in the profundal zone at a depth of 27 m and greater (Figure S2b).

### 3.5. The Effect of Depth on the Species Diversity of Testate Amoebae

Both the mean number of species per sample and the total number of species per sampling site (Figure 8a,b) varied with depth and showed similar drops and peaks. The number of species increased on the littoral reaching the greatest values on sublittoral, decreased at the bottom slope, and greatly varied in the profundal zone. The most considerable drops in the species number were detected at four depths of 18, 24, 42, and 57 m, whereas the greatest peaks were observed at the depths 20, 36, and 51–54 m. Both Shannon and Simpson diversity indices demonstrated the same patterns along the depth gradient (Figure 8c,d). There were considerable drops in diversity at the depth of 12–15 m (i.e., near the lower border of the thermocline). The values of indices decreased from 21 to 27 m, increased afterward to the depth of 36 m, then dropped at the depth of 42 m.

### 3.6. Optima and Tolerance of Testate Amoebae to Sampling Depth

Species *Difflugia minuta*, *D. oblonga*, *Cylindrifflugia elegans*, *C. acuminata*, and *Centropyxis cassis* were found in more than 75% of samples at all 20 of the sampling sites (Table S3) so that the distribution patterns along the depth gradient were visualized separately (Figure 9). The relative abundance of *D. oblonga* increased with depth, reaching a maximum of 38% of the total number of shells at the bottom slope (15 m). In deeper biotopes, the relative abundance decreased, but at depths of 24–30 m and 42 m, local peaks were recorded. The relative abundance of *D. minuta* increased from 0.2% on the littoral to 6.7% in the profundal zone (42 m), varying from 3 to 5% at depths deeper than 42 m. The maximum relative abundance of *C. acuminata* was observed on the littoral at a depth of 3 m only, whereas, at the other depths, the shells of this species occurred sporadically. The relative abundance of *C. elegans* did not exceed 4% on the littoral, increased to 14% in the sublittoral, then smoothly fluctuated around this value. The relative abundance of *C. cassis* increased with depth; local peaks were found at depths of 12 and 24 m; a sharp decrease was noted at the depths of 15 and 27–30 m.

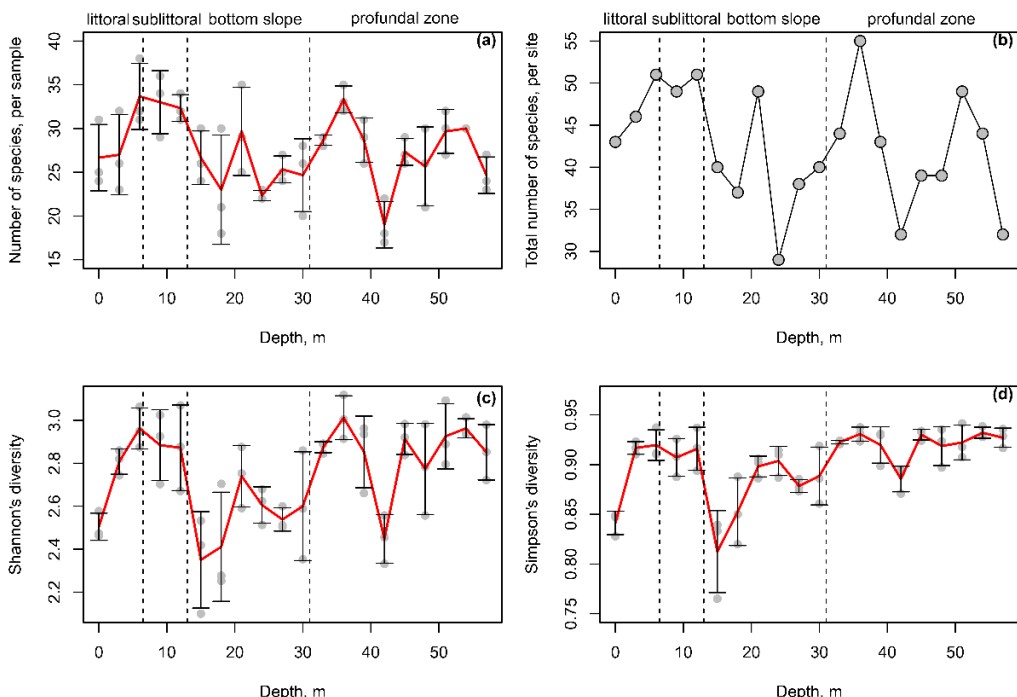

**Figure 8.** Mean number of species per sample (**a**) the total number of species per site (**b**), Shannon's (**c**), and Simpson's (**d**) diversity indices of testate amoeba assemblages along the depth gradient in Lake Valdayskoe. Bars are standard deviations; vertical dashed lines mark the borders between the depth zones.

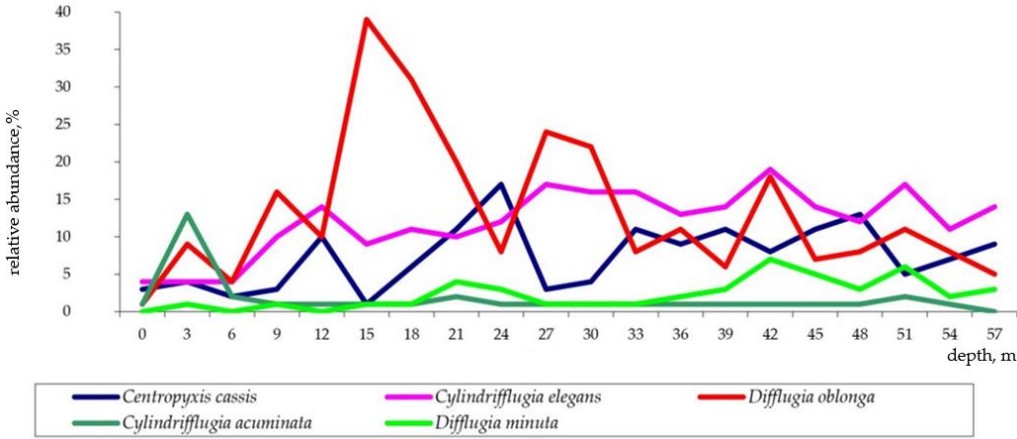

**Figure 9.** Relative abundance (%) of *Centropyxis cassis*, *Cylingifflugia acuminata*, *C. elegans*, *Difflugia minuta*, and *D. oblonga*.

The depth-related variation of relative abundances of eight species found in more than 75% of samples and at least at 18 stations (Table S3) is shown in Figure S3a,b. Relative abundance of *D. petricola*, reached peaks of 19.9% and 14.5% on the sublittoral at the depths 9 and 12 m, respectively, and then gradually decreased. *D. linearis* and *A. hemisphaerica* showed very similar patterns with the maximum relative abundances on the littoral (14% and 6%, respectively), very low abundance on the sublittoral and bottom slope, and a second slightly lower peak in the profundal zone at the depths deeper than 42 m. The relative abundance of *D. pulex* was low and increased slightly along the depth gradient from 0.2 to 3.9% with a maximum of 6.8% at 27 m. Distribution patterns of *D. minuta* were similar to those. The relative abundance of *D. penardi* increased from 1% on the littoral to 13% at the bottom slope (24 m) and varied from 5 to 10% starting from the depth of 27 m and deeper. The maximum relative abundance of *D. lithophila* (8%) and *D. longicollis* (4%)

was observed at the bottom slope (18–24 m) and slightly decreased in the profundal zone. *N. oviformis* demonstrated very low relative abundance in the profundal zone, where it increased to 9% and fluctuated between 6% and 9% to 57 m (with the only exception at a depth of 42 m, where the relative abundance was only 1%).

The species optimal depths and tolerance range are shown in Figure 10. The optimum depth continuously moves from 6 to 50 m depending on the species, but the majority of species can survive in a wide range of depths. It is possible to define the group of stenobathic and conditionally stenobathic species. We have considered species with a tolerance range of less than 10 m as stenobathic. These were *Difflugia ventricosa* (living in the depth range of 3–13 m), *D. claviformis* (9–12 m), *D. sinuata* (6–12 m), and *D. bicornis* (46–52 m). The conditionally stenobathic species demonstrated the manifested peak of relative abundance in a narrow depth range (less than 9 m) while at all the other depths they appeared sporadically with relative abundance. These were *Centropyxis aculeata*, *C. discoides*, *C. ecornis*, *C. laevigata*, *Galeripora discoides*, *Difflugia giganteacuminata*, and *D. rubescens* (Figure S4).

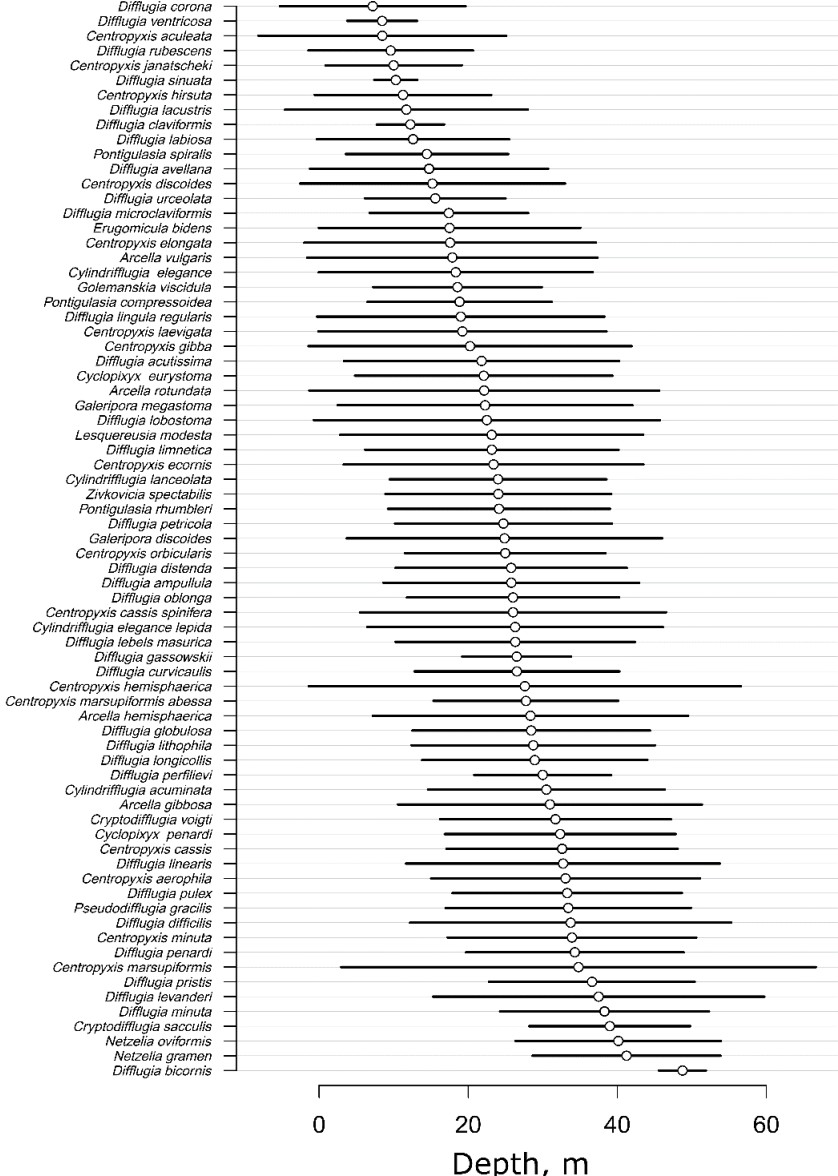

**Figure 10.** The depth optima and tolerance range of testate amoebae (only taxa encountered in three or more samples are shown).

## 4. Discussion

The obtained data allowed us to describe the fauna of benthic testate amoebae in Lake Valdayskoe for the first time, analyze changes in the assemblages of testate amoebae along the depth gradient, and evaluate the possibility of using testate amoebae as bioindicators of changes in the water depth in the past. The hydrological characteristics observed during this study correspond well to those collected over 72 years [29] which makes our data representative.

### 4.1. General Characteristics of Testate Amoeba Assemblages

The species structure of testate amoeba assemblages in Lake Valdayskoe is typical for benthic assemblages of freshwater lakes and is dominated by *Difflugia*, *Centropyxis*, *Arcella*, and *Cylindrifflugia*. These genera have been shown to be typical for benthic communities in freshwater lakes, as well as ecosystems in Europe, Asia, and North America [17,24,43–45]. We observed a relatively high abundance of two species of the genus *Netzelia* and three species of the genus *Pontigulasia*. The consistent presence of *Pontigulasia* species has been previously reported for testate amoeba assemblages in Čertovo and Laka lakes in Check republic [46] and in small lakes of New Brunswick and Nova Scotia [13]. The presence of *Netzelia* species was reported in the Bilé, in the Leopoldo and Pau Véio backwater [11], and in the Meshchera Lowlands [17]. We did not find any shells of the genus *Euglypha*, whereas *Nebela* was represented by only two tests on the transect. According to previous studies, species of these genera are often relatively abundant in lacustrine biotopes [44,47,48]. The absence of *Nebela* and *Euglypha* in the studied transect may be explained by hydrochemical conditions and requires further research. Thus, the benthic testate amoeba assemblages in the Lake Valdayskoe can be considered as typical for surface sediments of freshwater lakes and the observed differences from other reservoirs may be related to the hydrological features of the lake.

### 4.2. Variation in Species Structure of Testate Amoeba Assemblages along the Depth Gradient

The results of variation partitioning demonstrate that sampling depths and related environmental variables explained a considerable proportion (40.2%) of the total variance in the species structure of testate amoeba assemblages. The individual effect of the sampling depth accounted for 7.4% being the second most important factor after temperature which explained 19.2%. Similar results were previously reported by Cockburn et al. [49] who showed that the sampling depth explained 10.1% of the total variance in the species structure of benthic testate amoeba assemblages in Silver Lake (Eastern Ontario, Canada) and was the second most important factor after titanium (Ti) concentration, but the temperature accounted for 3.1% only. The more pronounced individual effect of temperature in our study could be related to a stronger thermocline in Lake Valdayskoe and deeper sampling depths. The effects of depth might be also related to the variation in substrate type and particle size composition as they were not added separately in the analysis. Previous studies [50] showed that the size structure of the mineral fraction in sediments greatly affected (31%) the species structure of testate amoeba assemblages. The individual effect of near-bottom water pH was relatively weak (1.5%), but when considered together with temperature and sampling depth (i.e., the overall effect) it considerably contributed to the variance (13.2%). The significant effects of water pH on the species structure of testate amoeba assemblages were previously shown by a number of studies [16,51,52]; however, in some environmental settings, like ours, it might be difficult to disentangle its effects from other factors. The same is true for the bottom inclination angle of which the individual effect is negligible, but the overall effect is about 6.2%. The influence of the bottom relief on testate amoebae has not been studied; however, studies on marine ciliates assemblages [53], marine invertebrates [54], and freshwater bacteria [55] have shown a noticeable impact of bottom relief on benthic assemblages. Overall, the sampling depth significantly affects the species composition of testate amoeba assemblages, mostly through the related temperature and pH of near-bottom water.

The changes in the species structure of testate assemblages along the depth gradient were clearly linked to bottom zones traditionally used in limnology—littoral, sublittoral, bottom slope, and profundal. The previous studies on the distribution of testate amoebae along the sampling depth gradient describe three types of benthic assemblages [12,17,24,49]. This difference can be explained by the covered depth ranges and the lake basin relief. Lake Valdayskoe is deeper (max depth 57 m) as compared to the other investigates sites Rabisha reservoir, Ovcharitsa Reservoir in Bulgaria, and the lakes in the Meshchera Lowlands where maximal depth ranged from 12 to 33 m. Indeed, the basin of Lake Valdayskoe consists of all the mentioned zones (littoral, sublittoral, bottom slope, and profundal), whereas smaller lakes might be without a bottom slope or, in some cases, sublittoral zones [29].

1.  Littoral Assemblages (LA). Our data indicate the dominance of species of the genus *Centropyxis*, *Cylindrifflugia*, and *Arcella* on the littoral that correlates to the observations in Ovcharitsa Reservoir [56], Smradlivo Ezero glacial lake [57], and the lake Dorian [58] in Bulgaria. In all these studies, the abundance of *Centropyxis* and *Arcella* at shallow depths was higher than at depths greater than 6 m. The high abundance of *Centropyxis aculeata*, *C. discoides*, and *Galeripora discoides* on the littoral corresponds well to the findings of Dabes & Velho [59], who showed the dominance of these species on the littoral in Lake São Francisco (Brazil). The confinement of *Cylindrifflugia. acuminata* and *Arcella hemisphaerica* to the conditions of the littoral were already shown in a study by Schönborn [60]. Sigala [61] reported the dominance of *Centropyxis aculeata* and *Galeripora discoides* in the sites with high oxygen content and warm water. Thus, the species structure of littoral assemblages in Lake Valdayskoe is similar to the littoral assemblages in other lakes.

2.  Sublittoral Assemblage (SA) considerably differs from the LA and is mainly formed by the species which were previously reported by many authors in oligo-, meso- and eutrophic lakes [49,62,63]. These authors also mention the high relative abundance of the genus *Difflugia*, the low relative abundance of *Centropyxis*, and the almost complete absence of *Arcella* at depths deeper than 9 m. *D. oblonga* was very abundant in the SA, but was not identified as indicator species for this assemblage. It may be explained by a high level of eutrophication and a thin sapropel layer in the sediments that create good conditions for the high reproduction of *D. oblonga* [11]. The list of identified indicator species may be divided into two groups. The first group is formed by *Difflugia urceolata*, *D. petricola*, and *Pontigulasia rhumbleri* which were previously observed only at depths greater than 5 m [13,57,64]. Another group of species includes *Zivkovicia spectabilis*, *Difflugia claviformis*, and *Golemanskia viscidula* which were previously reported in a depth range of 1.5–8 m [15,57] that also includes the shallow waters. Moreover, Torigai [65] claimed that *G. viscidula* depends on biotope productivity, but not on depth. Thus, the specimen structure of sublittoral assemblages consists of species earlier reported from the related or shallower depths.

3.  Bottom Slope Assemblage (BA) is described for the first time, so that there are no previous descriptions of assemblages of this type in the literature. Genus *Difflugia* dominates in BA, while genera *Cylindrifflugia* and *Centropyxis* occupy only 10–15% of the total abundance. There are three indicator species (*Difflugia oblonga*, *D. lithophila* and *Cyphoderia ampulla*) in this assemblage, however, *Difflugia oblonga* was usual in other assemblages and *D. lithophila* depends on high trophic status [66]. The most abundant species in the BA also have high relative abundance in sublittoral and profundal assemblages that might indicate a transitional nature of BA.

4.  Profundal assemblage (PA) was characterized by the presence of *Netzelia oviformis*, *N. gramen*, *Difflugia minuta*, and *D. pristis*. These findings correspond well with the results of Todorov [57] and Davidova [24], who described similar species structures in the profundal zone and mentioned *Netzelia oviformis*, *N. gramen*, and *Difflugia minuta* as typical species for profundal. Davidova [24] observed *Netzelia* species in benthic biotopes of the Rabisha reservoir (Bulgaria) with the maximal relative abundance at the deepest sites (12–15 m). Tsyganov et al. [17] mentioned *N. gramen* as a species-

typical for sublittoral and profundal at depths from 4.5 to 20.5 m in Shatura lakes (Moscow region, Russia). Arriera [11] described the positive correlation between *Netzelia oviformis* and the sampling depth. Thus, the profundal assemblage species structure is typical for deep water lacustrine habitats.

Analysis of dominance curves indicates that the littoral and bottom slope assemblages were characterized by steep dominance curves, i.e., the presence of a single dominant species, whereas the relative abundance of subdominants was two-free times lower. The sublittoral and the profundal assemblages were characterized by smoother dominance curves with one or three dominants and three or five subdominants with similar values of relative abundance. The dominance curve of assemblages is an efficient tool for estimation of the habitat stress level in a benthic assemblage [40]. Ivanov and Kosheleva [67] showed a positive correlation between the steepness of dominance curves and the stress level in habitats. The high-stress level in the littoral zone might be caused by the wave load, daily and annual temperature fluctuations, and the low content of available organic matter, while at the bottom slope the high stress may be a result of water flows and a large angle of inclination. The lower stress in the sublittoral and profundal zones might be related to more stable environmental conditions there, even though the conditions in the deepest biotopes are not really favorable for testate amoebae.

### 4.3. Relative Abundance of Testate Amoeba Genera along the Depth Gradient

Our results indicate decreasing relative abundances of the genus *Centropyxis* in the sublittoral zone in favor of the genus *Difflugia*. The simultaneous peak of the abundance of genus *Difflugia*, mainly related to *D. oblonga*, and a drop in the abundance of genus *Centropyxis* registered on the underwater terrace might be related to the higher level of eutrophication there in comparison with slopes [68]. This corresponds well to the results of Wiik [69], who described an increase in the relative abundance of *D. oblonga* ("oblonga") and a simultaneous decrease in the relative abundance of *A. vulgaris* and *Centropyxis aculeata* ("aculeata") in biotopes with a greater content of organic matter. The dominance of *Difflugia* in eutrophic habitats was also described by Velho [70] and Arrieira [11]. The relative abundance of these genera on underwater terraces and slopes might be affected by the water flows [71].

### 4.4. The Effect of Depth on the Species Diversity of Testate Amoebae

The minimal diversity of benthic testate amoeba assemblages (the number of species, Shannon and Simpson indices) was observed at the depth of 0 m. That corresponds well with the results of the previous research of benthic and plankton testate amoebae [17,24,61,72,73] and can be expected due to relatively unfavorable environmental conditions, i.e., a high wave load, low organic matter content in the soil, high sand content in the sediments, significant daily temperature fluctuations, etc. The diversity reaches the maximum values in sublittoral assemblages at depths of 9–12 m. These results are in line with those of Tsyganov [17] and Davidova [24] and might be explained by the optimal environmental conditions for testate amoebae, e.g., pH [51] and organic matter content [64]. The considerable drop in species diversity at the depth of 15 m might be caused by the thermocline. The effect of the thermocline was previously described on testate assemblages [13], and on chironomid assemblages [74]. In contrast with Davidova [24] and Celis [75], we do not detect the decline of diversity in deep waters. We assume that the relatively high oxygen level and availability of organic content in deep waters in Lake Valdayskoe may explain the high diversity there.

### 4.5. Optima and Tolerance of Testate Amoebae to Sampling Depth

The distribution of species' relative abundance along the depth, the species optima, and tolerance in relation to sampling depth allows us to describe preferences of testate amoebae and identify species that can be used as indicators. We have classified the species into two groups and three subgroups:

1. Eurybathic species. *D. oblonga* inhabits all depths with the greatest abundance in the biotopes with high eutrophication levels that corresponds well to the findings of Kihlman and Kauppila [76], and Arriera [11], who describe the positive correlation between *D. oblonga* and high levels of eutrophication. This suggests that *D. oblonga* is an indicator of high eutrophication, but not the depth, and is able to inhabit the depth gradient from 0 to 57 m. The absence of depth dependence for *Cylindiflugia elegans* was noted in the works of Davidova [15,24], who found *C. elegans* at all depths, from phytal to profundal. Tran [77] showed that *C. elegans* was a typical species in all studied aquatic habitats in Vietnam regardless of depth. We propose that *D. elegans* is a eurytopic species. The ability of *C. cassis* to inhabit all the depths also has been shown by Todorov [57] and Davidova [15].

2. Conditionally stenobathic species:

   a. Littoral species (0–6 m depth). The high abundances of *Cylindriflugia acuminata*, *Difflugia rubescens*, and *Arcella hemisphaerica* in the littoral correlate well with the findings of Schönborn [60] who also mentioned that these species as typical for shallow waters. Even more evidence for such ecological preferences can be found for *C. acuminata* [60] and *D. rubescens* [78], so these two species can be considered reliable indicators for littoral conditions. At the same time, our results indicate that the relative abundance of *Arcella hemisphaerica* slightly increased in profundal. We associate this with the intensive accumulation of the dead shells of *A. hemisphaerica* from plankton, where this species is also abundant. Therefore, this species should be used as an indicator of shallow waters with caution. In our study, *D. linearis* has maximal abundance in the littoral (the habitat with high content of the sand in sediments) which correlate well with the results of Golemansky [79], who mentioned that *D. linearis* is a psammophilic species in marine habitats without depth sampling specification. Littoral habitats are generally characterized by a high sand content, but further studies might be needed for a better estimation of the ecological preferences of this species. The same is true for *D. giganteacuminata* which was observed at depths from 5 to 40 m by Davidova [15]; however, without specification of the exact depths. In our study, we observed high abundances of *Centropyxis aculeata* on the littoral that correlates with the findings of Patterson et al. [13] who discover *C. aculeata* in shallow water at depths of less than 10 m. However, a subspecies *C. aculeata oblonga* was reported in deep water assemblages in Shatura Lakes by Tsyganov et al. [17]. This inconsistency makes us cautions against straightforward interpretations of the ecological preferences of this taxon. For the other mentioned in results species, *Centropyxis discoides*, *C. ecornis*, *C. laevigata* and *Galeripora discoides* there is a lack of information about their ecological preference, so further studies are needed for fill the gap in our knowledge.

   b. Sub-littoral species (6–15 m depth). Four species have a maximal abundance or depth optima in this zone: *D. ventricosa*, *D. claviformis*, *D. sinuata,* and *D. petricola*. The maximum relative abundance of *D. petricola* was observed at depths of 9–15 m, which corresponds to the results of Todorov [57], who discovered *D. petricola* only at depths greater than 5 m in the Smradlivo Ezero lake. *D. ventricosa* and *D. claviformis,* in contrast to our findings, were previously registered in littoral biotopes [15,18] with eutrophic and hypertrophic conditions. We assume that these species are distributed at depths of less than 15 m and require biotopes with high organic content and thin sediments. There are no data about the depth preferences of *D. sinuata*, so we cannot include or exclude this species from the list. Thus, *D. petricola* can be used as an indicator of depths between 5 m and 15 m and *D. ventricosa* and *D. claviformis* occur in a depth range from 0 to 15 m.

     c.    Deep-Water Species (>15 m). We found that six species *Difflugia penardi*, *D. minuta*, *D. lithophila*, *D. pulex*, *Netzelia oviformis*, and *Difflugia longicollis* had maximal abundance in the deep waters. Similar patterns were reported by Todorov [57] for *D. minuta* which was observed in the profundal zone; however, this study also reported that *D. pulex* inhabited not only the profundal zone, but also coastal mosses. This allows us to include *D. minuta* in the list of deep-depth indicators and excludes *D. pulex* from this list. Beyens et al. [80] reported *D. penardi* in biotopes with a pH of $6.52 \pm 0.8$, whereas in our study it was observed in a slightly acidic environment with a pH of 7.8–7.2 at depths of 24 m or deeper. Therefore, *D. penardi* may be considered an indicator of deep-water conditions. The maximum abundance of *D. lithophila* was observed at depths of 15–30 m. Macumber et al. [66] associated this species with the high trophic status of habitats but did not specify the sampling depth. There is no available data on the ecological preferences of *D. longicollis*. Thus, both species cannot now be set as deep-water indicators. The maximal relative abundance of *N. oviformis* in deep waters correspond well with the observations of Arrieira [11]. Although this species was also observed in plankton and periphyton, we believe that further researches allow us to prove the ability to use *N. oviformis* as a proxy of deep waters because the presence of testate amoebae in the plankton should not be considered stochastic [81]. Thus, *D. minuta* and *D. penardi* can be used as deep biotope indicators.

## 5. Conclusions

The structure of testate amoeba assemblages in bottom surface sediments and their distribution along the depth gradient in Lake Valdayskoe was described for the first time. The study showed that the benthic testate amoeba assemblages in the large and deep lake reflect the main ecological and limnology zones: littoral, sublittoral, bottom slope, and profundal. The dominance curve of assemblages reflects that the environmental conditions in the littoral and bottom slope are more changeable and affect the testate amoebae stronger than in the sublittoral and profundal. The species diversity was minimal in the extremal conditions at the depth of 0 m, and at the lower border of the thermocline. Ecological preferences of testate amoebae to the depth were identified and the eurybathic and conditionally stenobathic species were distinguished. This information may serve as a basis for the usage of benthic testate amoebae as a proxy in paleolimnological studies for inferring freshwater body depth. *Difflugia oblonga*, *Cylindrifflugia elegans*, and *Centropyxis cassis* were defined as eurybathic species, which do not depend on the depth itself but on the other environmental conditions, e.g., the level of eutrophication and pH. *Cylindriflugia acuminata*, *Difflugia rubescens*, and *D. linearis* mainly inhabit shallow waters; *D. ventricosa* and *D. claviformis* can be used as proxies for the depth range from 0 to 15 m; *D. petricola* is characteristic for the depths between 5 and 15 m; *D. minuta* and *D. penardi* are indicators of the deepest habitats. These data can be further used for the development of the "lake depth-testate amoeba assemblage" transfer function.

**Supplementary Materials:** The following supporting information can be downloaded at: https://www.mdpi.com/article/10.3390/d14110974/s1, Figure S1. Surface bottom sediments in the Birge-Ekman dredger; Figure S2. Relative abundance of testate amoeba genera along the depth gradient in Lake Valdayskoe; Figure S3. Relative abundance of *Netzelia oviformis*, *Difflugia linearis*, *D. pulex*, and *Arcella hemisphaerica* (a) and *Difflugia penardi*, *D. petricola*, *D. lithophila*, and *D. longicolis* (b) along the depth gradient in Lake Valdayskoe; Figure S4. Relative abundance of *Centropyxis ecornis*, *C. laevigata Difflugia giganteacuminata*, and *D. rubescence* (a) and *Centropyxis aculeata*, *C. discoides*, and *Galeripora discoides* (b) along the depth gradient in Lake Valdayskoe; Table S1. Characteristics of sampling sites along the depth gradient of Lake Valdayskoe; Table S2. The list of testate amoeba species, the number of counted shells, and their proportion in four assemblages along the depth gradient in Lake Valdayskoe; Table S3. The occurrence of testate amoebae species in the samples and on the stations along the depth gradient; Table S4. Relative abundance of genera in four assemblages along the depth gradient in Lake Valdayskoe.

**Author Contributions:** Conceptualization, V.V.S.; methodology, A.N.T., Y.A.M.; formal analysis, V.V.S., A.N.T.; investigation, V.V.S., F.Y.R.; data curation, V.V.S.; writing—original draft preparation, V.V.S.; writing—review and editing, V.V.S., A.N.T., Y.A.M.; visualization, V.V.S., A.N.T.; supervision, Y.A.M. All authors have read and agreed to the published version of the manuscript.

**Funding:** This research was funded by the Russian Science Foundation, grant number 19-14-00102. This research was performed according to the Development program of the Interdisciplinary Scientific and Educational School of M.V. Lomonosov Moscow State University "The future of the planet and global environmental change".

**Institutional Review Board Statement:** Not applicable.

**Data Availability Statement:** The data supporting reported results can be found in the Supplementary Materials to this paper.

**Acknowledgments:** Authors thank the Valday Branch of the State Hydrological Institute for the organization of the field works, the "Laboratory of water assemblages and invasions" (A.N. Severtsov Institute of Ecology and Evolution of Russian Academy of Sciences) for technical support of sample analysis, and Paul Bayley and Linda Bayley for proofreading.

**Conflicts of Interest:** The authors declare no conflict of interest. The funders had no role in the design of the study; in the collection, analyses, or interpretation of data; in the writing of the manuscript; or in the decision to publish the results.

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
