# Peer review of "The Effects of Sampling Depth on Benthic Testate Amoeba Assemblages in Freshwater Lakes: A Case Study in Lake Valdayskoe (the East European Plain)"

_diversity, doi:10.3390/d14110974_

Round 1

Reviewer 1 Report

Dear Authors,

this work is interesting and useful for the scientific community. This work is well structured and presented. However, changes must be incorporated before publication.

Abstract

1) Line 20: “The results of RDA”

You use the abbreviation RDA without indicating what it means, please mention the exact term

Introduction

2) Line 38: Briefly explain the term "proxy"

Material and methods

3) Line 98: in July 2021. (Figure 1b)

Put “(Figure 1b)” before the point. Please

Result

4) Please add information in the legend of your different figures (4 and 5). Specify what corresponds to what is found on the abscissa and the ordinate. Thanks

5) Table 1:  “Cylindrifflugia González-Miguéns et al., 2021........... »

Please, put the bibliographic references in the style suitable for the journal as it has been done in the text of your manuscript, thank you.

Discussion

6) Please remove the different titles for each new part in the discussion thank you.

7) Line 464: remove the space between waters and the point.

Author Response

Dear Revier 1 

thank you for revising the manuscript and your comments. Please see the attachment.

Reviewer 2 Report

There is a minor error, please correct it.

Author Response

Dear Reviewer 2

Thank you for your comments.

Reviewer 3 Report

The article seems to me to be a very interesting and well designed work. Here are some comments.

1. I think that the term "Bottom" can be confusing, since in other publications it is used to refer to the greater depth of a lake, therefore I suggest replacing it with "Surface sediment".

2. I consider it important to add to the description of the study site the data of the length and width of the lake.

3. In Field sampling first detail the total number of meters that the complete transect had. In Data analysis include the formula used to calculate the Shannon index.

4. In Results, they refer to acidity, but what they measured is pH and therefore it is better to refer to it as such (e.g. in Fig.3).

5. On line 271, I think it should say "fractions" not "factions".

6. On line 276, there is a parenthesis that does not close.

7. On line 315, it says the drop is in three depths and lists four.

8. In Discussion, there are a couple of publications that take a similar approach and find depth to be an important factor for testate amoebae (Sigala et al., 2018. Ecological drivers of testate amoeba diversity in tropical water bodies of central Mexico and Charqueño-Celis, 2019. Testate amoebae (Amoebozoa: Arcellinidae) as indicators of dissolved oxygen concentration and water depth in lakes of the Lacandón Forest, southern Mexico). 

  - In Table S2, the name of the authors goes before the varieties, and the varieties go without italics, e.g. change:    Centropyxis marsupiformis obesa Deflandre, 1929  by  Centropyxis marsupiformis Deflandre, 1929 obesa    - In Table S3, I think that the term "Share" is unclear, it could be replaced by "Frequency".

Author Response

Dear Reviewer 3.

Thank you for revising the manuscript. Please see the attachmment.

Reviewer 4 Report

Overall, the manuscript is very well written and relevant to the advancement of knowledge about the ecology of testate amoebae. The results presented by this work can also enrich the discussions of several other studies, such as those that address paleolimnological reconstruction, species replacement over time, eutrophication processes and the effects of climate change on aquatic ecosystems. The subject is interesting for the purpose of the journal and studies on testate amoebae are still rare, so it is a good contribution. So I have only two small comments about the manuscript.

1. You could add predictions about which assemblages of testate amoebae were expected at each depth;

2. Please consider the studies of Alves et al. to discuss your results, such as:

https://doi.org/10.1016/j.ejop.2011.10.004

https://doi.org/10.1016/j.ejop.2010.07.001

Author Response

Dear Reviewer 4

thank you for revising the manuscript. Please see the attachment.
